# The burden of catastrophic and impoverishing health expenditure in Armenia: An analysis of Integrated Living Conditions Surveys, 2014–2018

Jacob Kazungu[1], Christina L. Meyer[2]*, Kristine Gallagher Sargsyan[3], Seemi Qaiser[3], Adanna Chukwuma[3‡]

1 Health Economics Research Unit, KEMRI Wellcome Trust Research Programme, Nairobi, Kenya, 2 RTI International Center for Global Noncommunicable Diseases, Seattle, WA, United States of America, 3 Health, Nutrition, and Population Global Practice, World Bank Group, Washington, D.C., United States of America

‡ AC are joint senior author on this work.
* cmeyer@rti.org

**Data Availability Statement:** The data used in the analysis is publicly available online from at https://armstat.am/en/?nid=205 or the World Bank's

## Abstract

Armenia's health spending is characterized by low public spending and high out-of-pocket expenditure (OOP), which not only poses a financial barrier to accessing healthcare for Armenians but can also impoverish them. We analyzed Armenia's Integrated Living Conditions Surveys 2014–2018 data to assess the incidence and correlates of catastrophic health expenditure (CHE) and impoverishment. Households were considered to have incurred CHE if their annual OOP exceeded 40 percent of the per capita annual household non-food expenditure. We assessed impoverishment using the US$1.90 per person per-day international poverty line and the US$5.50 per person per-day upper-middle-income country poverty line. Logistic regression models were fitted to assess the correlates of CHE and impoverishment. We found that the incidence of CHE peaked in 2017 before declining in 2018. Impoverishment decreased until 2017 before rising in 2018. After adjusting for socio-demographic factors, households were more likely to incur CHE if the household head was older than 34 years, located in urban areas, had at least one disabled member, and had at least one member with hypertension. Households with at least one hypertensive member or who resided in urban areas were more likely to be impoverished due to OOP. Paid employment and high socioeconomic status were protective against both CHE and impoverishment from OOP. This detailed analysis offers a nuanced insight into the trends in Armenia's financial risk protection against catastrophic and impoverishing health expenditures, and the groups predominantly affected. The incidence of CHE and impoverishment in Armenia remains high with a higher incidence among vulnerable groups, including those living with chronic disease, disability, and the unemployed. Armenia should consider different mechanisms such as subsidizing medication and hospitalization costs for the poorest to alleviate the burden of OOP.

Microdata Library at https://microdata.worldbank.org/index.php/home.

**Funding:** The authors received no specific funding for this work.

**Competing interests:** The authors have declared that no competing interests exist.

## Introduction

As one of the key components of universal health coverage (UHC), financial risk protection in health aims to prevent households from exposure to financial hardship as a result of their direct health spending [1]. Since 2010, direct household out-of-pocket (OOP) payments still account for well over a third of low- and middle-income countries' (LMICs) total health spending, contributing to global financial hardship and impoverishment [2–4]. Globally, 996 million individuals experienced catastrophic health expenditures (CHE) out-of-pocket health spending that exceeds ten percent of a household's budget—while 435 million people were impoverished by OOP payments in 2017 [5].

Compared with other LMICs, Armenia has one of the highest ratios of OOP spending to total health expenditure, growing by 45 percent over a twelve-year period to reach 85 percent in 2019 [2]. As of 2018, Armenia's public health spending as a proportion of the country's gross domestic product (GDP) (1.2 percent) is much lower than other upper-middle income (UMI) countries (3.2 percent) and the average among countries in Europe and Central Asia (6.7 percent) [6]. Similarly, the country's public health spending as a proportion of the Armenian government's budget (5.2 percent) is lower than its European and Central Asian counterparts in 2018 (15.2 percent) [7]. The most recent examination of Armenia's CHE found that 16 percent of Armenian households' OOP payments exceeded 10 percent of annual household consumption in 2013 [3]. With an average annual 3.3 percent increase in the incidence of catastrophic OOP payments between 2010 and 2013, Armenia's incidence of catastrophic health payments has grown faster than any other country in the world [3].

Armenia provides public spending for services via an explicitly defined basic benefits package (BBP). In practice, there are three packages. The basic BBP package available to the whole population consists of outpatient services including primary care, maternity services, and sanitary epidemiological services. However, 38 percent of the population has access to the more extensive BBP package with inpatient service coverage, including the poor, vulnerable, and special groups. There is also a special package for civil and military servants [8]. Those who qualified for the more generous coverage, in 2006,paid 45 percent less during health care visits and had 36 percent higher rates of using outpatient care than those with less generous BBP coverage [9]. As of 2018, approximately 18 percent of Armenian households have at least one member with access to the BBP, including government workers and select socially vulnerable individuals [10].

As a reflection of the above, 62 percent of Armenians have to pay entirely out-of-pocket for outpatient diagnostic care, medication, and most inpatient care [8, 9]. The rest of the population (recipients of the more generous BBP coverage)have also required copayments for the same services, challenging efforts to reduce the disease and economic burden from Armenia's high prevalence of noncommunicable diseases [9, 11]. The cost of medication has historically been a major source of high OOP payments as the BBP pharmaceutical program coverage is limited, and pharmaceuticals are subject to a 20 percent value-added tax but not subject to other forms of pricing laws or regulations [8]. Moreover, because of the BBP's limited budget for provider reimbursements, healthcare providers that administer care to BBP patients often subsidize their revenue by increasing formal and informal fees for non-BBP eligible patients, leading to higher OOP spending in Armenia [8]. As a result, Armenia's OOP payments have been higher than other UMI countries as well as Armenia's Europe and Central Asia regional counterparts. In a recent survey, nearly a third of Armenians who had not used health care services in the last 12 months reported their primary barrier to care was their inability to pay for services [12].

Given the high OOP cost of care for Armenians, the impetus for this study is to understand the factors contributing to the country's financial barriers to health care access. Currently,

there is no comprehensive analysis of the incidence of OOP payments for health care in Armenia and limited data on CHE and impoverishing health spending after 2013. While the global literature indicates that: household economic status, the incidence of hospitalization, the presence of an elderly or disabled household member in the family, and the presence of a family member with a chronic illness are common significant factors associated with household CHE [5, 13], there are additional drivers of catastrophic and impoverishing health spending that are context-specific [14–21]. Among some European and Central Asian countries, the high cost of medication, outpatient coverage gaps, and physician-induced demand are cited as contributors to high OOP [22, 23].

This analysis aims to examine trends in the current state of Armenia's OOP expenditures and determine who is most at risk for CHEs or impoverishing health spending. It investigates sociodemographic factors, health status and service usage, and current benefit levels to understand their potential impact on catastrophic and impoverishing health expenditure in the Armenian context. By exploring the trends and inequalities in catastrophic and impoverishing health spending, this study both updates and adds nuance to the discussion around the current state of Armenia's financial risk protection and those most impacted by the cost of healthcare.

## Materials and methods

### Data sources

We obtained data from Armenia's Integrated Living Conditions Survey (ILCS) for five years from 2014 to 2018, within which the full complement of variables for the analysis was available and collected consistently [10]. However, the ILCS in Armenia has been conducted annually since 2001 by the National Statistical Service of the Republic of Armenia with support from the World Bank, United States Agency for International Development, and other donor organizations. The survey involves a two-stage stratified sampling approach where households are the ultimate sampling units. In this approach, the county is first stratified into sampling areas based on the location's geographical region, urbanization, and population size. Once stratified, eight households are randomly selected from each sampling area. Since its introduction, Armenia's ILCS has included a sample size of 5,184 households. However, the sample size was expanded to 7,872 between 2007 and 2011 following increased funding by the Millennium Challenge Account–Armenia (MCA-Armenia). The sample size reverted to 5,184 from 2012 except in 2017, where 7,776 households were included, because of budget availability. Ethical clearance for this study is deemed not to be required because it uses publicly available, de-identified data.

### Incidence of CHE and impoverishment

For each year between 2014 and 2018, we computed the incidence of CHE and impoverishment following widely applied methods [24, 25]. First, drawing on reported household spending, we calculated the total OOP expenditure incurred by patients while accessing care, summing individual expenditure at the household level. The OOP expenditure included spending on medicines, laboratory and radiology investigations, and other direct payments to the cashier or medical staff during inpatient or outpatient visits, including informal payments. However, the calculated OOP expenditure does not include the transportation costs to and from the health facilities, as this data is not available in the ILCS. We excluded any payments covered by health insurance. Cost variables for outpatient visits were based on a 30-day recall period. We annuitized these by multiplying the total OOP for outpatient visits by 12. However, inpatient costs were based on their last visit and a 12-month recall period. Therefore, we obtained total annual OOP costs for inpatient care by multiplying their respective inpatient

costs in the last visit with their total number of visits in the last 12 months. We imputed inpatient costs for those with inpatient visits but missing inpatient costs by multiplying the median day costs per admission by the average length of stay for admission in their last visit and the total number of visits in the last 12 months. We imputed zero for the length of stay for those who did not have inpatient visits to include the length of stay variable in the model. Then, we adjusted the outpatient, inpatient and total healthcare costs to 2018 constant Armenian Drams (AMD) using Consumer Price Index data from the World Bank Database.

CHE was measured by estimating the catastrophic headcount. A catastrophic headcount was defined as the percentage of households whose OOP expenditure for health care was greater than an established threshold. Two thresholds were used to assess the incidence of CHE. A household was considered to have incurred CHE if their per capita annual expenditure on health care exceeded: 1) 10 percent of the per capita total annual household consumption expenditure, or 2) 40 percent of the per capita annual household non-food consumption expenditure [26]. The two thresholds were used for comparison as a result of their differences in denominators (total household expenditure versus non-food expenditure), the threshold levels as a percentage of the denominator, and given the fact that there is no consensus about which of the two is the better threshold for assessing CHE [26, 27].

We assessed the incidence of impoverishment in Armenia through several steps. First, we defined the poverty line. We adopted two poverty lines for comparison: 1) the US$1.90 per person per day international poverty line and 2) the US$5.50 per person per day recommended for upper-middle income countries as suggested by the World Bank. Second, we used the GDP deflator data for Armenia to convert the poverty line to 2014, 2015, 2016, 2017, and 2018 values. Third, we converted this international poverty line for each respective year by multiplying by its respective World Bank purchasing power parity conversion factor (LCU per international dollar) into Armenian Drams (AMD). Finally, we calculated the percentage of households pushed further into poverty due to health expenditure by examining changes in total per capita expenditure before and after the household incurred health spending (standardized by household size).

## Inequalities in the distribution of CHE and impoverishing health expenditure

We computed the inequalities in the incidence of CHE and impoverishment using well-established methodologies described by Wagstaff et al. [28]. We calculated the rich-poor difference, defined as the difference in the percentage of households incurring CHE or facing impoverishment between the households in the richest quintile (Q5) to the poorest (Q1) and ratios (Q5/Q1). Although these two measures of inequality are easier to calculate and interpret, they do not consider poor (Q2), middle (Q3), and rich (Q4). Consequently, we generated the concentration curves and Wagstaff's concentration index for the CHE using both thresholds and impoverishing health expenditure for 2017 and 2018. The concentration curve plots indicated the cumulative percentage of CHE/impoverishment on the y-axis versus the cumulative population percentage ranked by wealth from poorest to richest [26]. When every household experiences the same share of CHE/impoverishment, the concentration curve transforms into a 45-degree line. CHE/impoverishment can be interpreted as concentrated among the rich when the curve lies below the 45-degree line and vice versa. The further the curve is from the 45-degree line, the higher the inequality. The concentration index is a summary measure obtained from the concentration curve and ranges from -1 to 1. A concentration index of 0 indicates equality in experiencing CHE/impoverishment, whereas the concentration index is a positive (negative) value when inequality is concentrated among the rich (poor) [29]. For this

analysis, the inequalities in impoverishment were calculated using the US$5.50 per person per day poverty line only as it is the recommended poverty line for upper-middle countries like Armenia.

## Correlates of CHE and impoverishing health expenditure

To assess the correlates of incurring CHE and impoverishment (using US$5.50 per person per day poverty line) from OOP expenditure on health care, we fit both bivariate and multivariable logistic regression models for each of the two outcomes: 1) whether a household incurred CHE at the 10 percent threshold (Yes/No) and 2) whether a household was pushed into poverty as a result of OOP expenditure on health (Yes/No). In each of these models, we incorporated sociodemographic factors including gender (Male/Female), age group (<25 years/25-34 years/45-54 years/55-64 years/65+ years), and level of education of the household head (None/ Primary/Secondary/Tertiary), whether household has at least one member with hypertension (Yes/No), whether at least one household member has private health insurance (Yes/No), whether at least one household member belongs to a group considered socially vulnerable or eligible for the social package (disabled, pensioner, receives social benefits, military, children), whether at least one household member has access to coverage for vulnerable groups under the BBP (Yes/No), whether at least one household member had some paid work (Yes/No), household location (Urban/Rural), household size (1–2 members/3-4 members/5+ members), and the household's socioeconomic status proxied by quintiles (poorest, poor, middle, rich, or richest quintile) of an asset index generated through principal component analysis. These independent variables were included in this analysis based on their relevance to the Armenian context (e.g., access to the BBP for vulnerable groups) or have previously been associated with the incidence of CHE and/or impoverishment in other studies [17–21, 24, 30]. Sampling weights were also applied to ensure that estimates were nationally generalizable while standard errors were clustered at the sampling area level. Additionally, we employed a complete case analysis where participants with missing data were excluded from the analysis. Overall, 98 percent of households had complete data on all variables included in the multivariable models.

## Inclusivity in global research

Additional information regarding the ethical, cultural, and scientific considerations specific to inclusivity in global research is included in S1 Checklist.

## Results

Table 1 presents the sociodemographic characteristics of the households surveyed in Armenia's 2014–2018 ILCS. Overall, the majority of surveyed households lived in urban areas; had elderly heads of household (65+ years); had at least one member with paid work; and were comprised of 3–4 household members. The proportion of female household heads in the study sample ranged from 31.4 percent in 2016 to 33.6 percent in 2017. Nearly all heads of surveyed households had a secondary education, half of whom received a tertiary education. Fewer than one out of ten surveyed households had one household member with hypertension. The proportion of households with a disabled member decreased from over 17 percent in 2014 to 14 percent in 2018. The poorest households made up a slightly larger share of surveyed households, ranging between 22.2 percent in 2017 and 24.4 percent in 2016.

Sampled households had varied access to governmental benefits or additional financial support. Whereas about half of surveyed households had at least one pensioner, fewer than one in ten sampled households had a member who received social benefits. By 2018, nearly one in five surveyed households had a member covered by the BBP for vulnerable groups, while just

**Table 1. Sociodemographic characteristics of surveyed households.**

| | 2014 | | 2015 | | 2016 | | 2017 | | 2018 | |
|---|---|---|---|---|---|---|---|---|---|---|
| | N | % | N | % | N | % | N | % | N | % |
| **Gender of household head** | | | | | | | | | | |
| Female | 1,635 | 31.54 | 1,706 | 32.91 | 1,625 | 31.35 | 2,610 | 33.56 | 1703 | 32.85 |
| Male | 3,549 | 68.46 | 3,478 | 67.09 | 3,559 | 68.65 | 5,166 | 66.44 | 3,481 | 67.15 |
| **Age group of household head** | | | | | | | | | | |
| <25 | 37 | 0.71 | 31 | 0.6 | 28 | 0.54 | 52 | 0.67 | 30 | 0.58 |
| 25–34 | 292 | 5.63 | 293 | 5.65 | 267 | 5.15 | 414 | 5.32 | 272 | 5.25 |
| 35–44 | 522 | 10.07 | 570 | 11 | 615 | 11.86 | 787 | 10.12 | 576 | 11.11 |
| 45–54 | 1,115 | 21.51 | 1,043 | 20.12 | 989 | 19.08 | 1,384 | 17.8 | 893 | 17.23 |
| 55–64 | 1,379 | 26.6 | 1,497 | 28.88 | 1,512 | 29.17 | 2,143 | 27.56 | 1,517 | 29.26 |
| 65+ | 1,839 | 35.47 | 1,750 | 33.76 | 1,773 | 34.2 | 2,996 | 38.53 | 1,896 | 36.57 |
| **Current marital status of household head** | | | | | | | | | | |
| Not Married | 1,948 | 37.59 | 1,924 | 37.11 | 1,889 | 36.44 | 2,967 | 38.16 | 1,932 | 37.27 |
| Married | 3,234 | 62.41 | 3,260 | 62.89 | 3,295 | 63.56 | 4,809 | 61.84 | 3,252 | 62.73 |
| **Level of Education of household head** | | | | | | | | | | |
| No Education | 40 | 0.77 | 35 | 0.68 | 24 | 0.46 | 44 | 0.57 | 16 | 0.31 |
| Primary | 691 | 13.33 | 616 | 11.88 | 594 | 11.46 | 825 | 10.61 | 535 | 10.32 |
| Secondary | 2,237 | 43.15 | 2,342 | 45.18 | 2,343 | 45.2 | 3,486 | 44.83 | 2,472 | 47.69 |
| Tertiary | 2,216 | 42.75 | 2,191 | 42.26 | 2,223 | 42.88 | 3,421 | 43.99 | 2,161 | 41.69 |
| **Whether at least one member has hypertension** | | | | | | | | | | |
| None | 4,696 | 90.59 | 4,651 | 89.72 | 4,640 | 89.51 | 7,138 | 91.8 | 4,782 | 92.25 |
| Yes | 488 | 9.41 | 533 | 10.28 | 544 | 10.49 | 638 | 8.2 | 402 | 7.75 |
| **Whether at least one member is disabled** | | | | | | | | | | |
| None | 4,279 | 82.54 | 4,290 | 82.75 | 4,308 | 83.1 | 6,473 | 83.24 | 4,439 | 85.63 |
| Yes | 905 | 17.46 | 894 | 17.25 | 876 | 16.9 | 1,303 | 16.76 | 745 | 14.37 |
| **Whether at least one member is a pensioner** | | | | | | | | | | |
| None | 2,585 | 49.86 | 2,647 | 51.06 | 2,577 | 49.71 | 3,769 | 48.47 | 2,556 | 49.31 |
| Yes | 2,599 | 50.14 | 2,537 | 48.94 | 2,607 | 50.29 | 4,007 | 51.53 | 2,628 | 50.69 |
| **Whether at least one member receives social benefits** | | | | | | | | | | |
| None | 4,886 | 94.25 | 4,899 | 94.5 | 4,935 | 95.2 | 7,218 | 92.82 | 4,931 | 95.12 |
| Yes | 298 | 5.75 | 285 | 5.5 | 249 | 4.8 | 558 | 7.18 | 253 | 4.88 |
| **Whether at least one member is in the military social group** | | | | | | | | | | |
| None | 5,125 | 98.86 | 5,105 | 98.48 | 5,109 | 98.55 | 7,640 | 98.25 | 5,109 | 98.55 |
| Yes | 59 | 1.14 | 79 | 1.52 | 75 | 1.45 | 136 | 1.75 | 75 | 1.45 |
| **Whether at least one member is in the children's social group** | | | | | | | | | | |
| None | 4,986 | 96.18 | 5,012 | 96.68 | 5,028 | 96.99 | 7,603 | 97.78 | 5,055 | 97.51 |
| Yes | 198 | 3.82 | 172 | 3.32 | 156 | 3.01 | 173 | 2.22 | 129 | 2.49 |
| **Whether at least one member has access to the BBP for vulnerable groups** | | | | | | | | | | |
| None | 4,274 | 82.45 | 4,171 | 80.46 | 4,151 | 80.07 | 5,971 | 76.79 | 4,238 | 81.75 |
| Yes | 910 | 17.55 | 1,013 | 19.54 | 1,033 | 19.93 | 1,805 | 23.21 | 946 | 18.25 |
| **Whether at least one member has health insurance** | | | | | | | | | | |
| None | 4,572 | 88.19 | 4,511 | 87.02 | 4,363 | 84.16 | 6,716 | 86.37 | 4,519 | 87.17 |
| Yes | 612 | 11.81 | 673 | 12.98 | 821 | 15.84 | 1,060 | 13.63 | 665 | 12.83 |
| **Whether at least one member has some paid work** | | | | | | | | | | |
| None | 1,286 | 24.81 | 1,302 | 25.12 | 1,327 | 25.6 | 1,952 | 25.1 | 1,233 | 23.78 |
| Yes | 3,898 | 75.19 | 3,882 | 74.88 | 3,857 | 74.4 | 5,824 | 74.9 | 3,951 | 76.22 |
| **Whether household is in an urban area** | | | | | | | | | | |

*(Continued)*

**Table 1.** (Continued)

| | 2014 | | 2015 | | 2016 | | 2017 | | 2018 | |
|---|---|---|---|---|---|---|---|---|---|---|
| | N | % | N | % | N | % | N | % | N | % |
| Rural | 3,348 | 64.58 | 3,348 | 64.58 | 3,348 | 64.58 | 5,184 | 66.67 | 1,944 | 37.5 |
| Urban | 1,836 | 35.42 | 1,836 | 35.42 | 1,836 | 35.42 | 2,592 | 33.33 | 3,240 | 62.5 |
| **Household size** | | | | | | | | | | |
| Small (1 to 2 members) | 1,511 | 29.15 | 1,607 | 31 | 1,594 | 30.75 | 2,580 | 33.18 | 1,777 | 34.28 |
| Average (3 to 4 members) | 1,776 | 34.26 | 1,847 | 35.63 | 1,844 | 35.57 | 2,632 | 33.85 | 1,829 | 35.28 |
| Bigger (5+ members) | 1,897 | 36.59 | 1,730 | 33.37 | 1,746 | 33.68 | 2,564 | 32.97 | 1,578 | 30.44 |
| **Household socioeconomic status** | | | | | | | | | | |
| Poorest | 1,227 | 23.67 | 1,202 | 23.19 | 1,264 | 24.38 | 1,704 | 22.19 | 1,222 | 24.06 |
| Poorer | 1,131 | 21.82 | 1,034 | 19.95 | 1,012 | 19.52 | 1,566 | 20.4 | 1,056 | 20.8 |
| Middle | 1,098 | 21.18 | 1,177 | 22.7 | 1,101 | 21.24 | 1,681 | 21.89 | 1,038 | 20.44 |
| Rich | 821 | 15.84 | 885 | 17.07 | 1,110 | 21.41 | 1,395 | 18.17 | 856 | 16.86 |
| Richest | 907 | 17.5 | 886 | 17.09 | 697 | 13.45 | 1,332 | 17.35 | 906 | 17.84 |

over a tenth of the surveyed households had a member with access to private health insurance. Less than two percent of surveyed households had a member in the military, while approximately three percent had a child who qualified as a part of the children's group [8].

## Health care costs–OOP payments

Table 2 shows the average and median household OOP payments for outpatient, inpatient, and total health care costs from 2014 to 2018 in constant 2018 Armenian Drams (AMD). Within each year, the average and median OOP payments for outpatient costs were higher than those for inpatient care. Overall, the mean OOP payments per household increased significantly from AMD 118,992 (95 percent confidence intervals (CI): 110,436–127,548) in 2014 to AMD 137,360 (95 percent CI: 119,581–155,140) in 2016 (p-value<0.009) and then dropped to AMD 124,733 (95 percent CI: 115,049–134,418) in 2017 (p-value = 0.209) and AMD 114,613 (95 percent CI: 98,235–130,992) in 2018 (p-value = 0.823). Between 2014 and 2018,

**Table 2. Trends in annual OOP payments in Armenia between 2014 and 2018 in Armenian Drams (AMD)– 2018 constant AMD.**

| | | Outpatient (AMD) | Inpatient (AMD) | Total health care costs (AMD) |
|---|---|---|---|---|
| **2014** | **N** | 5,184 | 5,184 | 5,184 |
| | **Mean [95% CI]** | 111,994 [103,611–120,378] | 6,998 [5,896–8100] | 118,992 [110,436–127,548] |
| | **Median [IQR]** | 19,057 [0–104,176] | 0 [0–0] | 22,938 [0–114,339] |
| **2015** | **N** | 5,184 | 5,184 | 5,184 |
| | **Mean [95% CI]** | 117,780 [98,100–137,460] | 6,695 [5,672–7,719] | 124,475 [104,723–144,227] |
| | **Median [IQR]** | 25,474 [0–122,474] | 0 [0–0] | 33,068 [0–122,474] |
| **2016** | **N** | 5,184 | 5,184 | 5,184 |
| | **Mean [95% CI]** | 128,201 [110,557–145,844] | 9,160 [7,739–10,580] | 137,360 [119,581–155,140] |
| | **Median [IQR]** | 26,914 [0–31,675] | 0 [0–0] | 124,217 [0–124,217] |
| **2017** | **N** | 7,776 | 7,776 | 7,776 |
| | **Mean [95% CI]** | 116,619 [107,067–126,171] | 8,115 [7,161–9,068] | 124,733 [115,049–134,418] |
| | **Median [IQR]** | 27,065 [0–123,24] | 0 [0–0] | 30,756 [0–123,24] |
| **2018** | **N** | 5,184 | 5,184 | 5,184 |
| | **Mean [95% CI]** | 108,425 [92,136–124,714] | 6,188 [5,211–7,165] | 114,613 [98,235–130,992] |
| | **Median [IQR]** | 22,200 [0–96,000] | 0 [0–0] | 24,000 [0–102,000] |

**Table 3. Trends in the incidence of CHE among Armenian households.**

| | Threshold—10% of total consumption | | | Threshold—40% of non-food consumption | | |
|---|---|---|---|---|---|---|
| | Percentage | 95% CI | | Percentage | 95% CI | |
| | | Lower | Upper | | Lower | Upper |
| 2014 | 19.86 | 18.65 | 21.13 | 8.50 | 7.67 | 9.42 |
| 2015 | 19.64 | 18.45 | 20.88 | 6.26 | 5.57 | 7.03 |
| 2016 | 19.50 | 18.32 | 20.73 | 6.77 | 6.06 | 7.56 |
| 2017 | 21.00 | 20.02 | 22.02 | 7.93 | 7.29 | 8.63 |
| 2018 | 18.71 | 17.55 | 19.93 | 5.53 | 4.87 | 6.27 |

the mean total OOP payments by households declined by 3.68 percent, with a larger share of this contributed by OOP payments for inpatient care, which declined by 11.57 percent while outpatient care also declined by 3.19 percent.

## Trends in the incidence of CHE and impoverishing health expenditure

Table 3 outlines CHE incidence at the 10 percent of total consumption and 40 percent of non-food expenditure thresholds. Similar trends in CHE are found upon examining both thresholds. At the 10 percent threshold, the incidence of CHE peaked in 2017, where 21.0 percent (95 percent CI: 20.02–22.02) of the population incurred CHE. On the other hand, the percentage of the population that incurred CHE is much lower each year when considering the 40 percent of non-food consumption threshold. For instance, the incidence of CHE was 7.93 percent (95 percent CI: 7.29–8.63) in 2017 using the 40 percent threshold. Irrespective of whichever threshold is used, over five percent of Armenians incurred CHE due to OOP payments for outpatient and inpatient care annually. Using the World Bank estimates of Armenia's population in 2020, this translates to over 148,162 individuals [31]. Regardless of the threshold, fewer Armenian households incurred CHE in 2018 compared to 2014. For instance, 47,928 (representing 171,004 Armenians) and 289,131 (representing 1,031,592 Armenians) fewer households incurred CHE in 2018 compared to 2014, when the 10 percent and 40 percent thresholds are considered, respectively.

Table 4 shows the results of the impoverishing effects of OOP payments using the US$1.90 per day international poverty line. Before incurring any health-related expenditure, 1.22 percent and 0.89 percent of Armenians lived below the international poverty line in 2014 and 2018, respectively. However, after spending on health, the poverty headcounts increased by 0.66 percent (95 percent CI: 0.45–0.97) (2014) and 0.70 percent (95 percent CI: 0.48–1.02) (2018), meaning that 19,300 and 20,662 Armenians were pushed into poverty in 2014 and 2018, respectively.

When applying the US$5.50 per day poverty line, there was a considerable shift in the proportion of households below the poverty line before and after spending on healthcare

**Table 4. Trends in the poverty headcount and poverty gap before and after OOP payments for health in Armenia between 2014 and 2018 using the US$1.90 per person per day international poverty line.**

| | Gross OOP health care payments | | | Net OOP health care payments | | | Difference | | |
|---|---|---|---|---|---|---|---|---|---|
| | Percentage | 95% CI | | Percentage | 95% CI | | Percentage | 95% CI | |
| | | Lower | Upper | | Lower | Upper | | Lower | Upper |
| 2014 | 1.22 | 0.94 | 1.58 | 1.88 | 1.52 | 2.33 | 0.66 | 0.45 | 0.97 |
| 2015 | 1.00 | 0.75 | 1.33 | 1.33 | 1.04 | 1.69 | 0.33 | 0.21 | 0.53 |
| 2016 | 0.68 | 0.49 | 0.96 | 1.27 | 0.98 | 1.64 | 0.58 | 0.39 | 0.87 |
| 2017 | 0.46 | 0.34 | 0.64 | 0.84 | 0.66 | 1.07 | 0.37 | 0.26 | 0.55 |
| 2018 | 0.89 | 0.66 | 1.19 | 1.59 | 1.26 | 2.01 | 0.70 | 0.48 | 1.02 |

**Table 5. Trends in the poverty headcount and poverty gap before and after OOP payments for health in Armenia between 2014 and 2018 using the US$5.50 per person per day poverty line.**

| | Gross of OOP healthcare payments | | | Net of OOP healthcare payments | | | Difference | | |
|---|---|---|---|---|---|---|---|---|---|
| | Percentage | 95% CI | | Percentage | 95% CI | | Percentage | 95% CI | |
| | | Lower | Upper | | Lower | Upper | | Lower | Upper |
| 2014 | 38.32 | 36.88 | 39.77 | 42.59 | 41.12 | 44.08 | 4.28 | 3.70 | 4.94 |
| 2015 | 34.92 | 33.52 | 36.34 | 39.52 | 38.07 | 40.99 | 4.60 | 4.01 | 5.28 |
| 2016 | 30.16 | 28.82 | 31.54 | 33.36 | 31.97 | 34.77 | 3.19 | 2.72 | 3.74 |
| 2017 | 30.46 | 29.36 | 31.58 | 36.00 | 34.85 | 37.17 | 5.54 | 5.00 | 6.13 |
| 2018 | 31.61 | 30.24 | 33.01 | 36.38 | 34.95 | 37.84 | 4.77 | 4.15 | 5.48 |

(Table 5). 38.32 percent and 31.61 percent of households were already below the poverty line before incurring any OOP expenditures for health in 2014 and 2018, respectively. After considering these expenditures, 4.28 percent and 4.77 percent more Armenian households were impoverished in 2014 and 2018, respectively.

Fig 1 reports the incidence of impoverishing effect of OOP payments for health care in Armenia between 2014 and 2018. The percentage of Armenians pushed further into poverty after health care payments fluctuated between 2014 and 2018. On average, when we applied the US$1.90 per person per day international poverty line, 5.63 percent more Armenian households (translating to 166,138 Armenians) were pushed into poverty because of OOP payments for health care in 2018 compared to 2014.

## Inequalities in the distribution of CHE and impoverishing health expenditure

Fig 2 shows the concentration curves for CHE at both the 10 percent of total consumption and 40 percent of non-food consumption thresholds from 2014 to 2018. Fig 3 shows the

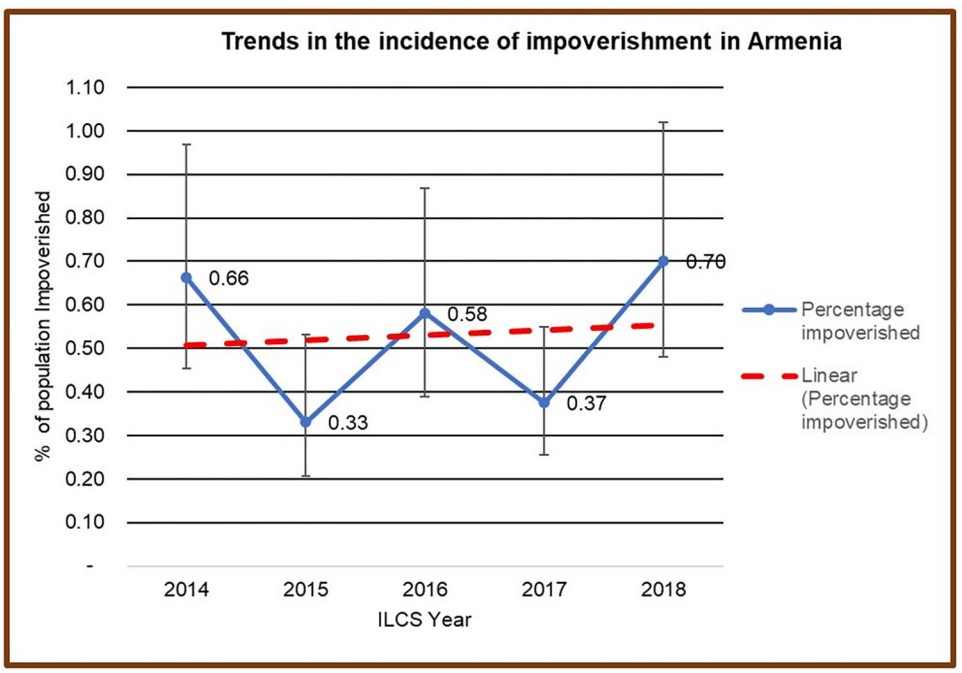

**Fig 1. Trends in impoverishment in Armenia between 2014 and 2018.**

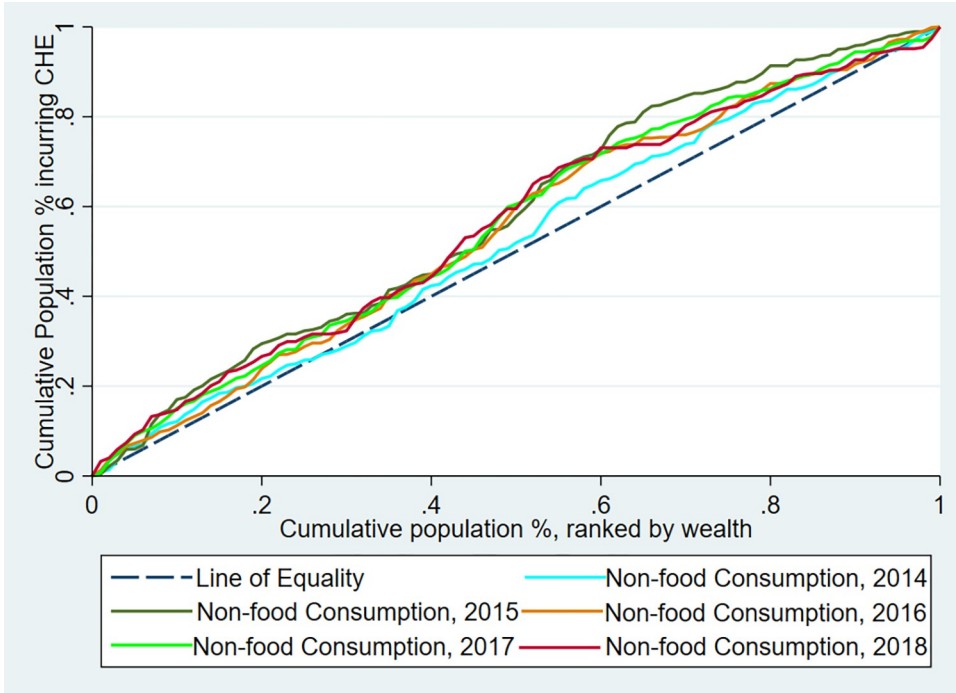

**Fig 2. Concentration curve for CHE by threshold and year in Armenia—2014 to 2018 ILCS.**

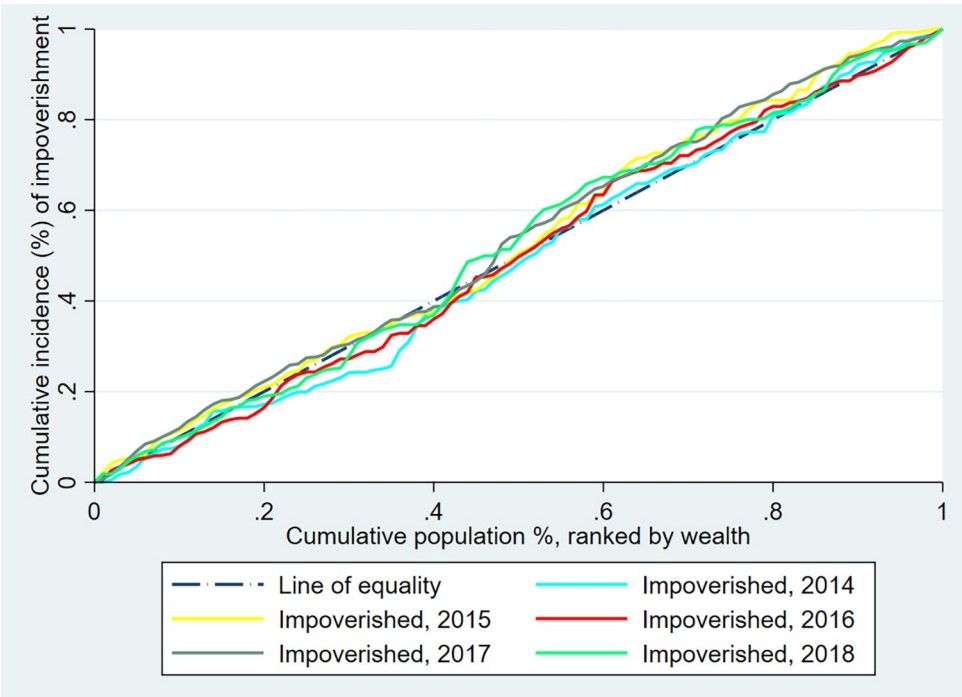

**Fig 3. Concentration Curve for impoverishment by year in Armenia– 2014 to 2018 ILCS.**

**Table 6. Inequalities in the incidence of CHE and impoverishment in the 2017 and 2018 ILCS in Armenia.**

| Year | Measure | | Q1 | Q5 | Difference (Q5-Q1) | High-to-low ratio (Q5/Q1) | Concentration index (CIX 95% CI) |
|------|---------|---|------|------|------|------|------|
| 2014 | CHE Thresholds | 10% of total consumption | 18.62 | 18.37 | -0.25 | 0.99 | 0.068*** (0.029 to 0.107) |
|      |          | 40% of non-food consumption | 9.3 | 6.61 | -2.69 | 0.71 | -0.051 (-0.108 to 0.005) |
|      | Impoverishment | | 3.6 | 4.3 | 0.70 | 1.17 | 0.254 (-0.025 to 0.103) |
| 2015 | CHE Thresholds | 10% of total consumption | 15.75 | 16.93 | 1.18 | 1.07 | -0.021 (-0.060 to 0.019) |
|      |          | 40% of non-food consumption | 4.87 | 2.7 | -2.17 | 0.55 | -0.170*** (-0.234 to -0.105) |
|      | Impoverishment | | 4.71 | 3.63 | -1.08 | 0.77 | -0.048 (-0.615 to -0.068) |
| 2016 | CHE Thresholds | 10% of total consumption | 18.13 | 17.96 | -0.17 | 0.99 | 0.011 (-0.028 to 0.051) |
|      |          | 40% of non-food consumption | 6.75 | 5.33 | -1.42 | 0.79 | -0.105** (-0.167 to -0.042) |
|      | Impoverishment | | 3.09 | 3.02 | -0.07 | 0.98 | -0.005 (-0.438 to -0.025) |
| 2017 | CHE Thresholds | 10% of total consumption | 21.25 | 18.37 | -2.88 | 0.86 | -0.018 (-0.050 to 0.014) |
|      |          | 40% of non-food consumption | 9.61 | 4.27 | -5.34 | 0.44 | -0.154*** (-0.202 to -0.106) |
|      | Impoverishment | | 6.08 | 3.49 | -2.59 | 0.57 | -0.067* (-0.124 to -0.011) |
| 2018 | CHE Thresholds | 10% of total consumption | 20.16 | 15.83 | -4.32 | 0.79 | -0.043* (-0.084 to -0.002) |
|      |          | 40% of non-food consumption | 7.51 | 2.89 | -4.62 | 0.39 | -0.170*** (-0.240 to -0.099) |
|      | Impoverishment | | 4.35 | 4.12 | -0.23 | 0.95 | -0.047 (-0.121 to 0.028) |

\* p-value<0.05

\*\*p-value<01

\*\*\*p-value<0.001

concentration curves for impoverishment between 2014 and 2018 in Armenia. Table 4 also presents the incidence of CHE in the poorest quintile (Q1), richest quintile (Q5), absolute differences (Q5-Q1), rich-poor ratios (Q5/Q1), and concentration index (CIX). Overall, there is a higher incidence of both CHE and impoverishing health expenditure among the poorest quintile compared to the richest quintile in Armenia. Specifically, in absolute terms, the incidence of CHE peaked in 2017, where the poorest experienced a 2.88 percent and 5.34 percent increase in CHE incidence compared to the richest at the 10 percent total consumption and 40 percent non-food consumption thresholds, respectively. The incidence then slightly declined in 2018. Based on the 40 percent of non-food expenditure threshold for CHE, the inequalities in CHE were significantly higher among the poorest relative to all other quintiles except in 2014; (CIX = -0.051, 95 percent CI: -0.108 to 0.005, p-value = 0.074) in 2014, (CIX = -0.170, 95 percent CI:-0.234 to -0.105, p-value<0.001) in 2015, (CIX = -0.105, 95 percent CI: -0.167 to -0.042, p-value = 0.001) in 2016, (CIX = -0.154, 95 percent CI: -0.202 to -0.106, p-value<0.01) in 2017 and (CIX = -0.170, 95 percent CI: -0.240 to -0.099, p-value<0.001) in 2018. Impoverishing health expenditure was also characterized by a higher incidence among the poor relative to the other quintiles across all years examined but was only significantly higher in 2017 (Table 6).

## Correlates of CHE and impoverishing health expenditure

Table 7 presents the unadjusted odds ratio (uOR), adjusted odds ratio (aOR), and p-values for the correlates of CHE health care in Armenia using the 2018 ILCS. The adjusted findings indicate that households headed by individuals older than 34 years were more likely to incur CHE than households headed by those under the age of 25 years. In addition, households with at least one person having hypertension were over five times (aOR = 5.20, 95 percent CI: 3.77–7.19, p-value<0.001) more likely to incur CHE compared to households where none of the members had hypertension. Furthermore, households with at least one member who belonged

**Table 7. Unadjusted and adjusted odds ratio for correlates of CHE.**

| | Catastrophic Health Expenditure (CHE) | | | |
|---|---|---|---|---|
| | Un-adjusted odds ratio [95% CI] | p-value | Adjusted odds ratio [95% CI] | p-value |
| **Gender of household head (Ref. Female)** | | | | |
| Male | **0.74 [0.59–0.94]** | **0.013** | 0.83 [0.58–1.18] | 0.296 |
| **Age group of household head (Ref. <25)** | | | | |
| 25–34 | 1.39 [0.59–3.27] | 0.449 | 1.98 [0.72–5.44] | 0.187 |
| 35–44 | **2.02 [1.49–2.74]** | **<0.001** | **2.87 [2.09–3.95]** | **<0.001** |
| 45–54 | **2.10 [1.45–3.03]** | **<0.001** | **2.58 [1.80–3.69]** | **<0.001** |
| 55–64 | **3.85 [3.05–4.86]** | **<0.001** | **3.83 [2.91–5.04]** | **<0.001** |
| 65+ | **5.75 [4.27–7.74]** | **<0.001** | **2.72 [1.76–4.22]** | **<0.001** |
| **Current marital status of household head (Ref. Not married)** | | | | |
| Married | **0.79 [0.64–0.98]** | **0.03** | 1.25 [0.88–1.77] | 0.205 |
| **Level of Education of household head (Ref. No education)** | | | | |
| Primary | .01 [0.57–7.15] | 0.278 | 1.55 [0.43–5.56] | 0.503 |
| Secondary | 1.55 [0.35–6.86] | 0.566 | 1.44 [0.36–5.77] | 0.61 |
| Tertiary | 1.70 [0.41–7.09] | 0.464 | 1.58 [0.42–5.95] | 0.499 |
| **Whether at least one member has hypertension (Ref. None)** | | | | |
| Yes | **6.17 [4.91–7.74]** | **<0.001** | **5.20 [3.77–7.19]** | **<0.001** |
| **Whether at least one member is disabled (Ref. None)** | | | | |
| Yes | **2.79 [2.30–3.38]** | **<0.001** | **2.12 [1.56–2.87]** | **<0.001** |
| **Whether at least one member is a pensioner (Ref. None)** | | | | |
| Yes | **2.63 [2.23–3.10]** | **<0.001** | 1.22 [0.92–1.63] | 0.169 |
| **Whether at least one member receives social benefits (Ref. None)** | | | | |
| Yes | 1.21 [0.93–1.59] | 0.162 | 0.74 [0.46–1.20] | 0.219 |
| **Whether at least one member is in the military social group (Ref. None)** | | | | |
| Yes | 1.34 [0.95–1.88] | 0.091 | 0.93 [0.71–1.23] | 0.623 |
| **Whether at least one member is in the children's social group (Ref. None)** | | | | |
| Yes | 0.59 [0.32–1.11] | 0.103 | 0.74 [0.31–1.78] | 0.503 |
| **Whether at least one member has access to BBP for vulnerable groups (Ref. None)** | | | | |
| Yes | **1.49 [1.16–1.91]** | **0.002** | 1.20 [0.96–1.50] | 0.114 |
| **Whether at least one member has health insurance (Ref. None)** | | | | |
| Yes | 0.77 [0.53–1.11] | 0.159 | 1.08 [0.69–1.67] | 0.746 |
| **Whether at least one member has some paid work (Ref. None)** | | | | |
| Yes | **0.33 [0.28–0.39]** | **<0.001** | **0.55 [0.47–0.64]** | **<0.001** |
| **Whether household is in an urban area** | | | | |
| Urban | **1.84 [1.15–2.95]** | **0.011** | **2.19 [1.03–4.65]** | **0.042** |
| **Household size (Ref. Small—1 to 2 members)** | | | | |
| Average (3 to 4 members) | **0.47 [0.39–0.57]** | **<0.001** | 0.80 [0.61–1.07] | 0.133 |
| Bigger (5+) | **0.47 [0.34–0.64]** | **<0.001** | 0.61 [0.35–1.09] | 0.096 |
| **Number of household members aged <18 years** | | | | |
| | **0.74 [0.66–0.84]** | **<0.001** | 1.01 [0.87–1.17] | 0.935 |
| **Number of household members aged 18 to 65 years** | | | | |
| | **0.77 [0.71–0.83]** | **<0.001** | 1.03 [0.88–1.21] | 0.721 |
| **Number of household members aged >65 years** | | | | |
| | **1.75 [1.57–1.96]** | **<0.001** | **1.47 [1.12–1.94]** | **0.006** |
| **Household socioeconomic status (Ref. Poorest)** | | | | |
| Poorer | 0.82 [0.63–1.08] | 0.157 | **0.60 [0.46–0.77]** | **<0.001** |
| Middle | 1.03 [0.78–1.37] | 0.843 | 0.70 [0.47–1.03] | 0.073 |

*(Continued)*

**Table 7.** (Continued)

| | Catastrophic Health Expenditure (CHE) | | | |
| --- | --- | --- | --- | --- |
| | Un-adjusted odds ratio [95% CI] | p-value | Adjusted odds ratio [95% CI] | p-value |
| Rich | 0.90 [0.60–1.35] | 0.608 | 0.62 [0.36–1.06] | 0.083 |
| Richest | 0.75 [0.49–1.14] | 0.187 | **0.52 [0.29–0.96]** | **0.036** |
| **N** | | | 5,078 | |
| **R²** | | | 0.1339 | |

to a disabled group (aOR = 2.12, 95 percent CI: 1.56–2.87, p-value<0.001) or located in urban areas (aOR = 2.19, 95 percent CI: 1.03–4.65, p-value = 0.042) were significantly more likely to incur CHE. Besides, every additional household member aged >65 years was associated with a 47 percent (aOR = 1.47, 95 percent CI: 1.12–1.94, p-value<0.001) increase in the odds of incurring CHE. However, households with at least one member engaged in some paid work had a 45 percent (aOR = 0.55, 95 percent CI: 0.47–0.64, p-value<0.001) reduced odds of incurring CHE after controlling for other factors. Similarly, better off households had lower odds of incurring CHE compared to poor households (Table 7).

Table 8 presents the uOR, aOR, and p-values for the correlates of impoverishing health care expenditures in Armenia using the 2018 ILCS.

Although the current marital status was not a significant factor in bivariate analyses, in the adjusted analysis, households with heads that were married had significantly 44 percent (aOR = 1.44, 95 percent CI: 1.05–1.99, p-value = 0.024) increased odds of being impoverished as a result of OOP payments. Additionally, households with at least one hypertensive member (aOR = 2.51, 95 percent CI: 1.56–4.03, p-value<0.001), at least one member belonging to the pensioner group (aOR = 1.94, 95 percent CI: 1.28–2.93, p-value = 0.002), at least one member belonging to the military social group (aOR = 2.17, 95 percent CI: 1.23–3.81, p-value = 0.007), or was located in urban areas (aOR = 1.76, 95 percent CI: 1.10–2.83, p-value = 0.019) were significantly more likely to be impoverished compared to households without hypertension, a member belonging to a pensioner or military social groups, or located in rural areas. Again, households with high socioeconomic status were protected from becoming impoverished because of OOP payments for health compared to households with a low socioeconomic status (Table 8).

## Discussion

This study presents the most recent comprehensive update of catastrophic and impoverishing health expenditures in Armenia since 2013. Moreover, it is one of the first studies to examine the correlates of Armenia's catastrophic and impoverishing health expenditures. Based on our analysis, Armenia's OOP payments declined between 2014 and 2018 due to declining outpatient and inpatient costs. Although there was a general decrease in CHE incidence among Armenian households between 2014 and 2018, CHE dramatically increased between 2013 and 2014, and in 2017 –coinciding with a fall in per capita public health spending between 2016 and 2017—before dipping again in 2018 [32]. Similarly, impoverishing health spending fell overall between 2014 and 2018, despite a peak in 2017.

The variation in CHE and impoverishment between 2014–2018 can be attributed to various reasons. First, over time, there have been changes to the benefits package composition, such as services, tariffs, and qualifying groups [33, 34]. For example, the maximum age for children eligible for more generous coverage under the BBP has continued to change, from three years old in 2001 to eighteen in 2019 [34]. Moreover, the BBP's service list differs over time as the

**Table 8. Un-adjusted and adjusted odds ratio for the correlates impoverishing health expenditure in Armenia—2018 ILCS.**

| | Impoverishment | | | |
| --- | --- | --- | --- | --- |
| | Un-adjusted odds ratio [95% CI] | p-value | Adjusted odds ratio [95% CI] | p-value |
| **Gender of household head (Ref. Female)** | | | | |
| Male | 1.15 [0.91–1.47] | 0.249 | 1.29 [0.90–1.85] | 0.163 |
| **Age group of household head (Ref. <25)** | | | | |
| 25–34 | **0.17 [0.06–0.50]** | **0.001** | 0.47 [0.13–1.70] | 0.251 |
| 35–44 | **0.26 [0.13–0.52]** | **<0.001** | 0.63 [0.30–1.31] | 0.216 |
| 45–54 | **0.29 [0.16–0.55]** | **<0.001** | 0.80 [0.50–1.30] | 0.371 |
| 55–64 | **0.52 [0.39–0.69]** | **<0.001** | 1.18 [0.84–1.65] | 0.338 |
| 65+ | - | - | - | - |
| **Current marital status of household head (Ref. Not married)** | | | | |
| Married | 1.14 [0.95–1.35] | 0.151 | **1.44 [1.05–1.99]** | **0.024** |
| **Level of education of household head (Ref. No education)** | | | | |
| Primary | 0.89 [0.11–7.41] | 0.913 | 0.52 [0.05–5.45] | 0.583 |
| Secondary | 0.81 [0.09–7.34] | 0.849 | 0.63 [0.05–7.22] | 0.710 |
| Tertiary | 0.75 [0.07–7.67] | 0.810 | 0.60 [0.05–7.79] | 0.694 |
| **Whether at least one member has hypertension (Ref. None)** | | | | |
| Yes | **3.45 [2.44–4.88]** | **<0.001** | **2.51 [1.56–4.03]** | **<0.001** |
| **Whether at least one member is disabled (Ref. None)** | | | | |
| Yes | **2.11 [1.63–2.73]** | **<0.001** | 1.01 [0.41–2.48] | 0.978 |
| **Whether at least one member is a pensioner (Ref. None)** | | | | |
| Yes | **3.51 [2.16–5.70]** | **<0.001** | **1.94 [1.28–2.93]** | **0.002** |
| **Whether at least one member receives social benefits (Ref. None)** | | | | |
| Yes | 1.55 [0.92–2.60] | 0.099 | 1.81 [0.65–5.03] | 0.254 |
| **Whether at least one member is in the military social group (Ref. None)** | | | | |
| Yes | **2.37 [1.42–3.95]** | **0.001** | **2.17 [1.23–3.81]** | **0.007** |
| **Whether at least one member is in the children's social group (Ref. None)** | | | | |
| Yes | 0.74 [0.32–1.71] | 0.488 | 0.85 [0.40–1.82] | 0.676 |
| **Whether at least one member has access to BBP (Ref. None)** | | | | |
| Yes | **1.52 [1.17–1.96]** | **0.001** | 1.42 [0.86–2.33] | 0.167 |
| **Whether at least one member has health insurance (Ref. None)** | | | | |
| Yes | **0.65 [0.42–1.02]** | **0.06** | 0.83 [0.49–1.44] | 0.514 |
| **Whether at least one member has some paid work (Ref. None)** | | | | |
| Yes | **0.41 [0.25–0.66]** | **<0.001** | 0.74 [0.53–1.02] | 0.068 |
| **Whether household is in an urban area** | | | | |
| Urban | **1.61 [1.04–2.49]** | **0.033** | **1.76 [1.10–2.83]** | **0.019** |
| **Household size (Ref. Small—1 to 2 members)** | | | | |
| Average (3 to 4 members) | **0.49 [0.27–0.90]** | **0.022** | 0.95 [0.47–1.92] | 0.881 |
| Bigger (5+) | **0.69 [0.53–0.91]** | **0.008** | 1.00 [0.50–2.01] | 0.998 |
| **Number of household members aged <18 years** | | | | |
| | **0.87 [0.76–1.00]** | **0.044** | 1.10 [0.92–1.32] | 0.291 |
| **Number of household members aged 18 to 65 years** | | | | |
| | **0.78 [0.68–0.91]** | **0.001** | 0.89 [0.72–1.10] | 0.281 |
| **Number of household members aged >65 years** | | | | |
| | **2.11 [1.64–2.72]** | **<0.001** | 1.31 [0.95–1.80] | 0.105 |
| **Household socioeconomic status (Ref. Poorest)** | | | | |
| Poor | 1.10 [0.71–1.73] | 0.664 | 0.93 [0.64–1.35] | 0.72 |
| Middle | 1.58 [0.91–2.74] | 0.105 | 1.20 [0.83–1.75] | 0.334 |

*(Continued)*

**Table 8.** (Continued)

| | Impoverishment | | | |
|---|---|---|---|---|
| | Un-adjusted odds ratio [95% CI] | p-value | Adjusted odds ratio [95% CI] | p-value |
| Rich | 0.79 [0.57–1.10] | 0.164 | **0.63 [0.44–0.90]** | **0.011** |
| Richest | 0.94 [0.57–1.55] | 0.820 | 0.75 [0.44–1.30] | 0.309 |
| N | - | | 5,049 | |
| $R^2$ | - | | 0.0926 | |

list is primarily informed by recommendations from Armenia's Minister of Health. Thus, variable coverage levels of services and populations may in part explain changes in OOP and the incidence of CHE and impoverishment. Second, Armenia's wealth has increased over time. For instance, the Gross National Income (GNI) per capita (international US dollars) has increased by 26 percent, from US$ 10,490 in 2014 to US$ 13,230 in 2018 [35].

The increases in wealth translated to growth in household consumption. Monthly adult consumption increased overall between 2014 and 2018 from 47,622 AMD to 48,575 AMD–an annual increase of 24 percent [36]. The rise in consumption expenditure and parallel falls in OOP, translated to reductions in catastrophic health spending. In addition, the proportion of the population living below the UMI poverty line of $5.50 a day decreased between 2014 and 2016 by seven percent before trending upwards in 2017 (increase of four percent) [37]. Despite the overall falls in poverty and rise in household consumption in the study period, 2017 was marked by a decline in public health spending that shifted the burden of health care to households and led to a rise in impoverishing and catastrophic health spending. The improvement in poverty rate and increasing consumption over the studied period strengthens the argument that OOP payments are the main driver of increasing CHE. In other words, despite the increasing prosperity enjoyed by Armenians, it was insufficient in protecting households against the impoverishing effects of CHE. Despite efforts by the government to target vulnerable groups with generous coverage, the incidence of catastrophic and impoverishing health expenditures was disproportionally concentrated among Armenia's more vulnerable socioeconomic groups between 2014 and 2018. Examining the impoverishing health expenditure concentration curves and index values demonstrates that the poorest Armenians are much more likely to be pushed into poverty from health spending than any other wealth quintiles. These disparities have also grown over time. While the wealthiest were less likely to experience CHE and impoverishment in 2018 compared to 2014, Armenia's poorest were more likely to experience CHE and impoverishment in 2018 than in 2014 (i.e., more people in the lowest quintile experienced CHE and impoverishment in 2018 compared to 2014). The concentration of financial barriers to health care access among vulnerable groups suggests a need to increase or better target public spending for health benefits. Given ongoing efforts to target health benefits, these inequalities may also highlight the importance of universality in designing health benefits, which may improve equity and reduce inefficiencies arising from the cost of implementing targeting mechanisms. Our findings on the concentration of CHE and impoverishment among the poor are similar to those reported by other studies [3, 18, 38, 39].

Besides a household's level of wealth, a significant predictor of CHE was whether a household member had hypertension, a common risk factor for chronic noncommunicable diseases. After accounting for all other factors, households with a hypertensive member were more than five times as likely to incur a CHE. Households with elderly over the age of 65 were also 47 percent more likely to experience a CHE than households without any elderly members, even if they could receive some form of social benefits. These findings are not just reflective of global trends but were expected as elderly and individuals with chronic conditions often tend to have

more facility visits than young and healthy individuals [5]. Our findings point to the importance of strengthening financial protection for people living with NCDs in Armenia, which is essential to prevent the development of complications that are relatively more expensive to manage, and productivity losses due to premature death and preventable disability. Similarly, households with a disabled member had more than twice the odds of incurring CHE compared to households without any disabled members, regardless of their benefits or insurance status. Interestingly, in contrast to other countries' experiences [40–42], urban residents had a greater odds of incurring CHE than their rural counterparts controlling for all other factors, including participation in social benefit programs. This may be explained by the fact that urban residents tend to bypass primary health care providers for expensive specialist care in the urban polyclinic model in which the scope of care of specialists and family physicians overlaps [43].

A critical protective factor against CHE was whether a household had at least one member with paid work. Regardless of one's level of wealth or other characteristics, households with at least one member with paid work had 45 percent reduced odds of incurring CHE compared to households without paid work. The presence of paid work—informal or formal—may be a proxy for a higher ability to pay for health services, buttressing the fact that the poorest households continue to incur a disadvantage in terms of access to care.

As for the impoverishing impact of health spending, urban households, households with members belonging to the military social group, households with at least one pensioner, and households with a hypertensive patient had higher odds of IHE. Although households with a disabled member were more likely to face impoverishment, these odds were not statistically different from their non-disabled counterparts. Surprisingly, access to social benefits or Armenia's BBP did not appear significantly impact the odds of experiencing impoverishing health expenditures, indicating Armenia's benefits package does not offer a sufficient level of support to meaningfully protect Armenians from impoverishing health spending. Hence, there is a need to re-examine the depth and service scope of coverage in the BBP in Armenia to ensure that it confers adequate financial risk protection to all Armenians.

Armenia has made strides to expand services covered under the BBP, including providing primary health services for the general population as well as inpatient services for the poor and other vulnerable groups. However, with Armenia's growing burden of chronic noncommunicable diseases and the high cost of outpatient and diagnostic treatment for most, the country's current health system and BBP cannot address the growing disparities in catastrophic and impoverishing health spending. Similar to other European and Central Asian countries, outpatient medications are a key driver of OOP payments in Armenia [44, 45]. Without pharmaceutical pricing regulation, the VAT tax on medication, and a negative perception of generic drugs among patients and providers, Armenia's reliance on OOP payments to finance expensive outpatient treatment puts many households at financial risk [45].

Increasing the government's financing for health would allow the government to strategically finance more health services with prepaid public resources [46]. For example, in a study of Eastern European countries, Estonia not only has the second highest public spending on health and second lowest levels of OOP payment but also offers a generous package that covers most hospital care with cost-sharing elements for pharmaceuticals, dental care, and therapeutic appliances [47, 48]. Several considerations may inform the options for financing an expanded benefits package in Armenia, including the population age structure, formal employment rates, the strength of tax administrative mechanisms, and the broader fiscal policies [5, 9, 31].

The study has several key limitations. The analysis of Armenia's ILCS demonstrates correlations between factors and outcomes and does not identify any causal relationships between

specific factors and the levels of health expenditures. The analysis cannot directly ascertain the individual circumstances that lead to differing household healthcare spending and garner further study. Similarly, while the authors conducted a literature review of common correlates of OOP payments, CHE, and IHE, the study may be at risk of omitted variable bias if there are other context-specific variables that we did not account for in our analysis. Given the survey's focus on Armenia, the results of our analysis may not apply to other contexts outside of Armenia.

This study did not include direct non-medical costs, such as transportation, due to data unavailability, which have proven to increase the incidence of CHE and IHE in other settings [18, 39]. Lastly, although examining healthcare expenditures precludes an analysis of those who may not seek care, analyzing the burden of catastrophic and impoverishing health expenditures provides insight into how cost affects healthcare use broadly amongst Armenians, which may also have implications for those who do not seek out healthcare. Future studies should include and explore these factors.

## Conclusion

This study offers a detailed examination of the burden of catastrophic and impoverishing health spending in Armenia to date. As the analysis draws from a nationally representative survey of Armenia, the results are generalizable throughout the population, providing a detailed picture of what Armenians are likely to experience on average. Investigating the association between socioeconomic factors and healthcare spending trends in Armenia provides greater insight into the progress Armenia has made on healthcare spending and which groups need more robust support to reduce the incidence and intensity of CHE and IHE. Considering nearly a fifth of Armenian households experienced a CHE in 2018, Armenia's efforts to expand UHC and offer patients greater financial risk protection may require the country to undertake reforms to increase prepaid pooled financing to finance an expanded benefits package for the whole population.

## Supporting information

**S1 Checklist. Inclusivity in global research.**
(DOCX)

## Acknowledgments

The study authors sincerely appreciate the Statistical Committee of the Republic of Armenia for access to the ILCS surveys.

## Author Contributions

**Conceptualization:** Jacob Kazungu, Adanna Chukwuma.

**Formal analysis:** Jacob Kazungu.

**Methodology:** Jacob Kazungu, Adanna Chukwuma.

**Supervision:** Adanna Chukwuma.

**Visualization:** Jacob Kazungu.

**Writing – original draft:** Jacob Kazungu, Christina L. Meyer.

**Writing – review & editing:** Jacob Kazungu, Christina L. Meyer, Kristine Gallagher Sargsyan, Seemi Qaiser, Adanna Chukwuma.

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
