## [Decision Letter · Decision Letter 0]

10 Mar 2022

PGPH-D-21-01115

The burden of catastrophic and impoverishing health expenditure in Armenia: An analysis of Integrated Living and Conditions Surveys, 2014-2018

Dear Dr. Meyer,

Thank you for submitting your manuscript to PLOS Global Public Health. After careful consideration, we feel that it has merit but does not fully meet PLOS Global Public Health’s publication criteria as it currently stands. Therefore, we invite you to submit a revised version of the manuscript that addresses the points raised during the review process.

We look forward to receiving your revised manuscript.

Kind regards,

Hassan Haghparast Bidgoli

Academic Editor

JOURNAL REQUIREMENTS:

Additional Editor Comments (if provided):

Reviewers have raised some concerns in particular in regards to the focus of the study and aligning the messages with the analysis, as well as methodology. On methodology, in particular, the following major issues

require more attention :

- the endogeneity issue and selection process of the explanatory variables for the analysing determinant of CHE and impoverishment

- reason for using asset index instead of consumption expenditure in creating quintiles

- appropriateness of the poverty threshold used for this setting

Reviewers' comments:

Reviewer's Responses to Questions

**Comments to the Author**

1. Does this manuscript meet PLOS Global Public Health’s publication criteria? Is the manuscript technically sound, and do the data support the conclusions? The manuscript must describe methodologically and ethically rigorous research with conclusions that are appropriately drawn based on the data presented.

Reviewer #1: Yes

Reviewer #2: Partly

2. Has the statistical analysis been performed appropriately and rigorously?

Reviewer #1: Yes

Reviewer #2: No

3. Have the authors made all data underlying the findings in their manuscript fully available (please refer to the Data Availability Statement at the start of the manuscript PDF file)?

Reviewer #1: Yes

Reviewer #2: No

4. Is the manuscript presented in an intelligible fashion and written in standard English?

Reviewer #1: Yes

Reviewer #2: Yes

5. Review Comments to the Author

Reviewer #1: This manuscript presents information on CHE and impoverishment for Armenia. The analysis is appropriate, if only standard. I do not generally see the point of bivariate analysis, but am not generally bothered by the analysis.

However, I find the introduction to be out-of-touch with the actual research, while the abstract and conclusion also offer discussion that seems unrelated to the analysis. Roughly speaking, I would like to see more effort go into the broad literature related to this sort of analysis. SDGs and thoughts on a program that have not really been implemented, at least by the time the data was collected, are not really appropriate.

For example, in the abstract, "Preventing the burden of catastrophic ... , and other health reforms." Yet, the analysis offers no control on the relevant program. Thus, the conclusion is complete speculation, based on other literature. In that regard, the analysis simply does not tie to the "story" in the paper, and that absolutely needs to change.

The first paragraph in the intro is about SDGs and pooling. Yet, the analysis is not. Thus, this is not particularly relevant. The focused discussion about Armenia (second paragraph) seems a better place to start, as it is directly about the actual paper. However, in that paragraph, there is a sentence, "... - found that 16% of OOP payments exceeded 10% of annual household consumption..." It does not come through cleanly, maybe something about 16% of households?

In the fourth paragraph, "To better address..," the one sentence that seems to justify this research, simply does not have enough support. The sentence is, "Reforms to address financial barriers to health ... determinants of these barriers." Although I understand the sentiment, I do not see how finding that age or access to a job has much to offer in terms of reform. I think this is a complaint I have with this entire literature and even the UHC dream. The barriers are broadly supply-based, while these determinants are mostly demand-based, although since they are after use has been observed, one could make the argument that these are equilibrium outcomes. Either way, they miss those that do not seek care for a variety of reasons, which seems to be far more relevant?

My point is not that the analysis is wrong, but I think the research is broadly oversold as having policy potential. I think it should instead be focused on what changes have been observed over this short period and whether or not there seem to be broad improvements, and, if not, can we point to any "groups" that seem to have been short-changed? That does not require even much change to the analysis. Rather, it requires a change to the focus.

Reviewer #2: This is an interesting paper looking at the incidence of financial hardship due to out of pocket payments in Armenia. It has great potential but I have a number of questions and comments:

“Using logistic regression models, data from 25 Armenia’s Integrated Living Conditions 2014 – 2018 was used to assess the incidence of catastrophic 26 health spending and impoverishment.” – logistic regression models are assessing the risk, not the incidence. Incidence is just a descriptive statistic.

The description of coverage policy and spending levels is useful for providing context, but could go more into depth. When did the BBP begin? More could be said about the depth of coverage.

Inpatient spending is imputed for some households with no reported inpatient spending but reported inpatient visits. However aren’t some inpatient services free at the point of use for some households? Could the authors please clarify in the text how they distinguish non-reporting of expenditure from free services? Also for how many households (ie what % of households with inpatient spending) were results imputed?

Headcount was said to be early on calculated as the percentage of households. But then later (e.g. line 205) the authors refer to % of population or even (line 210) the number of individuals. Which is correct (ie household weights or population weights)?

“The two thresholds were used 134 for comparison and given the fact that there is no consensus about which of the two is the better 135 threshold for assessing CHE (15).” It is not only the thresholds that make these methods different but also the denominator. See Cylus, Thomson, Evetovits (2018) “Catastrophic health spending in Europe: equity and policy implications of different calculation methods” Bulletin of the WHO for analysis that compares financial protection methods, including these two.

Are the quintiles used based on income, consumption or assets in the inequalities analysis? Are they adjusted for household composition? Do the results change using different definitions of household wealth (i.e. defining quintiles differently)?

Why do the models not also use 40% non-food as a dependent variable since it is calculated and presented in tables?

I have a number of concerns about the explanatory variables used in the models, many of which are endogenous. For example, access to BBP and being socially vulnerable. In the adjusted models, having access to BBP is not associated with more or less risk of catastrophic spending. This underscores the difficulties in interpretation with so many endogenous variables in a model.

But the same is true with education and socioeconomic status. This would seem to make it impossible to interpret the odds ratios in the adjusted models (and indeed, even in the unadjusted models due to lots of confounding)

Is hypertension a suitable proxy for health status? Are there other health variables in the dataset? Hypertension may also correlate with age, which itself correlates with more health care use.

Likewise being in paid work is also predicted by health status, as is high socioeconomic status. I am just not sure what the unadjusted models are telling you which would be useful for understanding policy.

Do the models control for year or are all years pooled without year fixed effects? I would think there could be variability in determinants over time, and indeed, the authors have shown results by year in the tables.

I’m also not sure about the impoverishign spending models given the very limited variability. Have the authors considered looking at spending for the “further impoverished” ie those who are below the poverty line and spending anything as well? This is becoming more common, even as part of global monitoring. Is $1.90 per person per day really appropriate for an upper middle income setting? One might argue for a higher poverty threshold (e.g. 60% median income even)?

Line 44 “as” should be “has”

Line 65: “found that 16 percent of OOP payments exceeded 10 percent of annual household consumption in 2013” is unclear. I think you mean that 16% of people lived in households for whom OOP payments exceeded 10 percent of annual household consumption?? Likewise the following sentence that catastrophic spending incidence has increased by 3.3% per year does not specify what years it is referring to.

What explains variability in CHE and impov over time?

No descriptive statistics are reported for the sample. What % are eligible for BBP, for example?

Where are medicines? Outpatient spending? Medicines are typically the driver of financial hardship.

“CHE incidence dramatically 323 increased in 2017 - reflecting in part a fall in per capita public health spending between 2016 and 2017 324 - before dipping again in 2018.” Line 323… this is not part of the analysis as far as I can tell.

6. PLOS authors have the option to publish the peer review history of their article (what does this mean?). If published, this will include your full peer review and any attached files.

**Do you want your identity to be public for this peer review?** For information about this choice, including consent withdrawal, please see our Privacy Policy.

Reviewer #1: No

Reviewer #2: No

---

## [Decision Letter · Decision Letter 1]

21 Jun 2022

PGPH-D-21-01115R1

The burden of catastrophic and impoverishing health expenditure in Armenia: An analysis of Integrated Living and Conditions Surveys, 2014-2018

Dear Dr. Meyer,

Thank you for submitting your manuscript to PLOS Global Public Health. After careful consideration, we feel that it has merit but does not fully meet PLOS Global Public Health’s publication criteria as it currently stands. Therefore, we invite you to submit a revised version of the manuscript that addresses the points raised during the review process.

We look forward to receiving your revised manuscript.

Kind regards,

Hassan Haghparast Bidgoli

Academic Editor

Journal Requirements:

1. Please update your online Competing Interests statement. If you have no competing interests to declare, please state: “The authors have declared that no competing interests exist.”

2. We have noticed that you have uploaded Supporting Information files, but you have not included a list of legends. Please add a full list of legends for your Supporting Information files after the references list. 

Additional Editor Comments (if provided):

Reviewers' comments:

Reviewer's Responses to Questions

**Comments to the Author**

1. If the authors have adequately addressed your comments raised in a previous round of review and you feel that this manuscript is now acceptable for publication, you may indicate that here to bypass the “Comments to the Author” section, enter your conflict of interest statement in the “Confidential to Editor” section, and submit your "Accept" recommendation.

Reviewer #1: All comments have been addressed

Reviewer #3: (No Response)

2. Does this manuscript meet PLOS Global Public Health’s publication criteria? Is the manuscript technically sound, and do the data support the conclusions? The manuscript must describe methodologically and ethically rigorous research with conclusions that are appropriately drawn based on the data presented.

Reviewer #1: Yes

Reviewer #3: Partly

3. Has the statistical analysis been performed appropriately and rigorously?

Reviewer #1: Yes

Reviewer #3: No

4. Have the authors made all data underlying the findings in their manuscript fully available (please refer to the Data Availability Statement at the start of the manuscript PDF file)?

Reviewer #1: Yes

Reviewer #3: Yes

5. Is the manuscript presented in an intelligible fashion and written in standard English?

Reviewer #1: Yes

Reviewer #3: Yes

6. Review Comments to the Author

Reviewer #1: Thank you for addressing our comments. I think the paper is now clearer in terms of matching what it does with what is recommended and discussed, and, therefore, am comfortable with it.

Reviewer #3: See attached file

7. PLOS authors have the option to publish the peer review history of their article (what does this mean?). If published, this will include your full peer review and any attached files.

**Do you want your identity to be public for this peer review?** For information about this choice, including consent withdrawal, please see our Privacy Policy.

Reviewer #1: No

Reviewer #3: **Yes: **Ajay Mahal

---

## [Editor Report · Decision Letter 2]

12 Sep 2022

The burden of catastrophic and impoverishing health expenditure in Armenia: An analysis of Integrated Living and Conditions Surveys, 2014-2018

PGPH-D-21-01115R2

Dear Ms. Meyer,

We are pleased to inform you that your manuscript 'The burden of catastrophic and impoverishing health expenditure in Armenia: An analysis of Integrated Living and Conditions Surveys, 2014-2018' has been provisionally accepted for publication in PLOS Global Public Health.

Best regards,

Hassan Haghparast Bidgoli

Academic Editor
